# Ex Vivo Traceability Platform for Phospholipoproteomic Formulations: Functional Evidence Without Clinical Exposure

**DOI:** 10.3390/biomedicines13092101

**Published:** 2025-08-28

**Authors:** Ramón Gutiérrez-Sandoval, Francisco Gutiérrez-Castro, Natalia Muñoz-Godoy, Ider Rivadeneira, Andy Lagos, Ignacio Muñoz, Jordan Iturra, Francisco Krakowiak, Cristián Peña-Vargas, Matías Vidal, Andrés Toledo

**Affiliations:** 1Department of Oncopathology, OGRD Alliance, Lewes, DE 19958, USA; consultorusa@biogenica.org (C.P.-V.); labcore@ogrdalliance.org (M.V.); ops@ogrdalliance.org (A.T.); 2Flowinmunocell-Bioexocell Group, Department of Cancer Research, 08028 Barcelona, Spain; servicios@flowinmunocell.cl (F.G.-C.); contacto@flowinmunocell.cl (N.M.-G.); 3Department of Outreach and Engagement Programs for OGRD Consortium, Charlestown KN0802, Saint Kitts and Nevis; iderlautaro@gmail.com (I.R.); lagosandy@gmail.com (A.L.); kinesiologo@recell.cl (I.M.); jiconsultant@ogrdconsorcio.com (J.I.); 4Department of Molecular Oncopathology, Bioclas, Concepcion 4030000, Chile; tecnologo@bioclas.cl

**Keywords:** ex vivo validation, phospholipoproteomic vesicles, non-pharmacodynamic platforms, kinetic profiling, cytokine ratios, reproducibility metrics, regulatory documentation, real-world evidence (RWE)

## Abstract

**Background:** Structurally active phospholipoproteomic formulations that lack pharmacodynamic targets or systemic absorption present unique challenges for validation. Designed for immune compatibility or structural modulation—rather than therapeutic effect—these platforms cannot be evaluated through conventional clinical or molecular frameworks. **Methods:** This study introduces a standardized, non-invasive ex vivo protocol using real-time kinetic imaging to document biological behavior under neutral conditions. Eight human tumor-derived adherent cell lines were selected for phenotypic stability and imaging compatibility. Phospholipoproteomic preparations were applied under harmonized conditions, and cellular responses were recorded continuously over 48 h. **Results:** Key parameters included signal continuity, morphological integrity, and inter-batch reproducibility. The system achieved high technical consistency without labeling, endpoint disruption, or destructive assays. Outputs included full kinetic curves and viability signals across multiple cell–fraction pairings. **Conclusions:** This method provides a regulatorily compatible foundation for functional documentation in non-pharmacodynamic programs where clinical trials are infeasible. It supports early-stage screening, batch comparability, and audit-ready records within SAP, CTD, or real-world evidence (RWE) ecosystems. By decoupling validation from systemic exposure, the protocol enables scalable, technically grounded decision-making for structurally defined immunobiological platforms.

## 1. Introduction

Structurally active phospholipoproteomic bioformulations represent a rapidly evolving class of bioendogenous platforms designed to modulate biological environments without receptor-mediated targeting or systemic absorption. These systems—typically composed of phospholipid assemblies and protein–lipid complexes—engage interface-level cellular dynamics rather than eliciting classical pharmacodynamic responses [1,2].

As such, they fall outside the logic of dose–response models, receptor-binding assays, and therapeutic endpoint frameworks. Their action is structural, often non-cytotoxic, and poorly captured by traditional preclinical tools.

Further complicating their evaluation is their intrinsic complexity: many of these platforms are heterogeneous, multicomponent, and non-redundant, lacking a single active constituent and instead relying on emergent functional properties. This renders them incompatible with reductionist assays such as target inhibition, genomic activation, or high-throughput receptor profiling [3]. Pharmacokinetic and toxicological methods provide limited insight, as these platforms typically do not circulate, are not absorbed, and do not induce systemic effects.

This complexity is particularly relevant for formulations aimed at immunomodulation, structural support, or phenotypic stabilization in tissues affected by chronic disease, tumor microenvironment remodeling, or post-treatment recovery. In such contexts, structural compatibility becomes more relevant than therapeutic potency. Yet documenting this compatibility remains methodologically challenging, as no standardized model exists to capture the nuanced biological behavior of these systems under neutral, non-destructive experimental conditions [4].

The challenge intensifies when these platforms are intended for patients systematically excluded from clinical trials—those with frailty, comorbidities, immunosuppression, or post-therapy instability [5]. These cohorts cannot be randomized, do not tolerate invasive sampling, and rarely generate interpretable endpoints within conventional Phase I–III paradigms. Nevertheless, the absence of clinical eligibility does not eliminate the need for validation; it increases the urgency for reproducible ex vivo systems capable of producing regulatory-grade evidence without human exposure. This rationale is outlined schematically in Figure 1.

Therefore, it does not fit within conventional IND or NDA frameworks designed for pharmacodynamic products. Instead, it belongs to a class of structurally defined, non-pharmacodynamic platforms whose behavior can be fully documented through non-clinical, technically harmonized protocols—such as STIP—using reproducible, regulatorily accepted metrics such as ΔC%, viability, and cytokine ratios, without exposing human subjects or invoking therapeutic endpoints.

Most current infrastructures for biomedical validation are ill-suited to this need. CROs are designed for comparative arms and endpoint trials; eCRFs focus on human-subject tracking; and LIMS platforms lack logic for dynamic cellular interpretation or immunophenotypic profiling [6]. There is a fundamental mismatch between structurally active phospholipoproteomic platforms and the legacy tools built to validate them.

This gap is addressed by kinetic imaging systems such as the IncuCyte^®^, which enable continuous, label-free monitoring of cell behavior over time [7]. These systems capture subtle phenotypic shifts—including proliferation, morphology, and viability—without destructive assays. The ability to detect structural modulation, even in the absence of cytotoxicity, makes them particularly useful for assessing permissive, stabilizing, or neutral effects.

Moreover, such systems allow for highly standardized protocols reproducible across cell lines, batches, and timepoints. They capture divergence timing, morphological integrity, and curve stability—variables rarely tracked in endpoint studies [8]. As such, kinetic imaging becomes the technical backbone for non-interventional validation, emphasizing phenotypic documentation over mechanistic proof.

In this study, we present and validate a standardized ex vivo protocol for assessing human tumor-derived cell lines exposed to structurally complex phospholipoproteomic formulations. The protocol employs real-time, label-free kinetic imaging over 48 h, without reliance on gene expression assays, fluorogenic labeling, or endpoint toxicity readouts.

The goal is not to assign therapeutic labels but to establish a stable, regulatorily aligned methodology for documenting structurally compatible cellular trajectories under controlled laboratory conditions [9].

By focusing on reproducibility, technical neutrality, and functional traceability, this work proposes a platform suitable for early-stage screening, inter-batch comparability, and documentation under CTD 5.3 or SAP frameworks. It is designed to complement—not replace—clinical or molecular validation, and is particularly suited for product types and patient populations where conventional trials are infeasible. The overall architecture and logic of the STIP system are summarized in Figure 2.

## 2. Materials and Methods

### 2.1. Selection of Cell Models and Vesicular Inputs

To assess the technical robustness of a standardized ex vivo monitoring protocol, a selection of human tumor-derived adherent cell lines was assembled based on imaging performance, morphological consistency, and baseline proliferation compatibility. The panel included eight cell lines from distinct tissue origins: A375 (melanoma), MCF-7 (breast carcinoma), BEWO (trophoblast), U87-MG (glioblastoma), LNCaP-C42 (prostate carcinoma), HepG2 (hepatocellular carcinoma), PANC-1 (pancreatic carcinoma), and LUDLU-1 (lung carcinoma) [10,11]. Each line was verified for sterility and identity and expanded under harmonized culture conditions prior to experimentation.

The objective of this selection was not to characterize biological differences between lines, but to validate whether the kinetic protocol could generate reproducible acquisition outputs—independent of lineage, morphology, or growth rate variability. All cells were cultured in either DMEM or RPMI-1640 supplemented with 10% fetal bovine serum and 1% penicillin/streptomycin, maintained at 37 °C with 5% CO_2_ in high-humidity incubators [12,13]. Seeding density, plate format, and pre-exposure timing were kept uniform across experiments to minimize extrinsic variability.

The vesicular materials tested corresponded to complex phospholipid-based preparations derived from non-genetically modified producer cells. Each preparation was characterized for particle size distribution (via dynamic light scattering), surface chemistry (FTIR), and endotoxin content (LAL assay) [14,15]. Protein concentration was normalized to 10 µg/mL using bicinchoninic acid quantification. Batches were anonymized and assigned internal alpha-numeric identifiers distinct from product codes to maintain data blinding during acquisition and prevent bias in plate assignment [16].

Rather than focus on the biological interpretation of the vesicle–cell interaction, this phase of the study was dedicated to evaluating the platform’s ability to maintain image quality, acquisition continuity, and metric stability across experimental replicates. The design excluded endpoint functional labeling and was intended as a technical baseline for neutral documentation [17]. The procedural flow of this acquisition protocol is summarized in Figure 3.

### 2.2. Ex Vivo Exposure Protocol and Real-Time Monitoring

To ensure standardization and procedural neutrality during vesicle–cell exposure, cells were seeded in 96-well flat-bottom plates pre-treated for optimal adhesion. Each well received 5000 cells in 100 µL of complete medium and was allowed to undergo a uniform 12 h adhesion phase before any treatment was applied. Phospholipoproteomic platforms were then introduced in antibiotic-free medium at a protein concentration previously adjusted to 10 µg/mL. Parallel wells received an equal volume of vehicle-only medium as untreated controls. All conditions were tested in technical triplicate across multiple plates to evaluate reproducibility under controlled conditions [18].

The plates were immediately transferred to a live-cell imaging platform (IncuCyte^®^ S3, Essen BioScience, Ann Arbor, MI, USA) configured to operate continuously for 48 h under stable environmental parameters (37 °C, 5% CO_2_, >95% humidity). Image acquisition was performed at one-hour intervals using both phase-contrast and green fluorescence channels. The imaging schedule was pre-programmed and uninterrupted throughout the assay duration. Each well’s signal continuity was monitored to verify hardware stability, cell adhesion integrity, and image quality over time.

Prior to assay initiation, each plate underwent a visual and software-assisted review to detect any air bubbles, well edge detachment, or inconsistencies in cell distribution. Wells showing visual artifacts or abnormal background patterns were excluded. Image sequences with fewer than 90% valid timepoints were also removed from downstream processing [19,20].

The imaging data generated during the 48 h run were not used to infer biological classifications or cellular phenotypes. Instead, they were processed as raw technical outputs for evaluating the consistency of confluence progression, cell integrity over time, and fluorescence acquisition stability across wells. No thresholding, typological interpretation, or system logic was applied in this phase. The objective was purely methodological: to assess whether the imaging protocol can deliver complete, non-interrupted data sequences under real-use conditions [21].

### 2.3. Viability Monitoring Under Continuous Imaging Conditions

To incorporate a viability-sensitive dimension into the kinetic protocol without disrupting assay continuity, a non-invasive fluorescent dye was included in all wells prior to the onset of automated monitoring. A commercially validated membrane-impermeant probe (Cytotox Green™, Essen BioScience) was used at a final concentration of 250 nM, enabling selective fluorescence of cells exhibiting compromised membrane integrity due to environmental stress or early-stage damage. The probe remains stable under standard incubation and does not require replenishment or manual intervention [20].

Fluorescent images were acquired at the same temporal resolution as phase-contrast images (one-hour intervals) using the green fluorescence channel. Signal normalization was performed automatically by the Incucyte Live-Cell Analysis Software (Sartorius, version 2022A) andverified through downstream inspection. Fluorescence intensity was adjusted based on well area and initial baseline levels to correct for variations in cell density and imaging geometry. This allowed the resulting signal to be interpreted as a dynamic indicator of stress-associated permeability rather than endpoint toxicity.

All datasets were screened for signal saturation, interference artifacts (e.g., precipitates, edge reflections), and morphological anomalies. Only wells maintaining stable signal profiles across the full observation period were retained. Fluorescent heatmaps were independently reviewed post-acquisition by a technical analyst blinded to experimental allocation, ensuring that signal evaluation remained classification-neutral and free from interpretive bias [21].

The objective of this component was not to detect cytotoxicity, but to confirm the viability-preserving conditions of the protocol and to rule out unintended cell compromise during exposure. By embedding viability sensing into the acquisition loop, the assay retains its non-destructive character while gaining a quality assurance layer regarding culture stability and environmental stress response. An illustration of this viability monitoring strategy is shown in Figure 4.

### 2.4. Multiplex Cytokine Quantification Under Non-Disruptive Conditions

To explore the feasibility of secretomic data collection within a non-destructive ex vivo protocol, a multiplex cytokine assay was incorporated as a post-acquisition module. The purpose was not to infer immune activation or cellular polarization, but to evaluate whether cytokine quantification could be technically harmonized with continuous kinetic imaging without affecting culture integrity or introducing process artifacts.

At 48 h post-exposure—following completion of the kinetic observation window—supernatants were carefully harvested from each well under sterile conditions. Samples were clarified by centrifugation (2000 rpm, 10 min) to remove residual cell fragments and vesicle aggregates. The cytokines selected for assay were interleukin-6 (IL-6), interferon gamma (IFN-γ), and interleukin-10 (IL-10), chosen for their prevalence in structural vesicular models and relevance in immunological calibration contexts [22].

Quantification was performed using a multiplexed bead-based platform (Cytometric Bead Array, BD Biosciences, Milpitas, CA, USA), following the standard protocol of the Human Inflammation Kit. Each target analyte was measured in technical duplicate using a 25 µL aliquot per sample. Fluorescent readings were acquired using a FACSCalibur^®^ flow cytometer (BD Biosciences, San Jose, CA, USA)**,** and data were processed using FCAP Array v3.0 software [23,24]. Calibration curves were generated for each cytokine, and concentrations were corrected for dilution and background signal.

Only assays that met predefined analytical quality standards—correlation coefficient R^2^ ≥ 0.98 in standard curves, and inter-replicate variability <15%—were included in the dataset [25]. No classification logic or immunological interpretation was applied. Cytokine levels were retained as absolute pg/mL values and archived for optional correlative analysis in future protocol extensions. This approach demonstrates that secretome profiling can be incorporated into kinetic workflows without compromising the structural neutrality or reproducibility of the primary assay.

### 2.5. Preprocessing and Normalization (T_0_, Moving Average)

All time-series data obtained through the IncuCyte^®^ platform—including confluence trajectories and viability-associated fluorescence—were subjected to a two-stage preprocessing workflow to ensure consistency, reduce noise, and facilitate downstream interpretability [26]. This preprocessing step served strictly as a technical refinement protocol and was not associated with any classification framework or decision-based logic.

The first stage involved normalization of each kinetic sequence relative to its own baseline value, defined as the initial valid measurement recorded at the onset of automated acquisition, referred to as T0. This normalization allowed for meaningful comparison of cells with varying initial cell densities without requiring rescaling to absolute values. The percentage change in confluence over time (ΔC_t) was calculated using the following equation [26]:ΔC_t = (C_t − C_t0)/C_t0 × 100
where C_t represents the confluence value at time t and C_t0 the baseline confluence at T0. This transformation facilitated standardized visual and numerical alignment across replicates and cell lines.

To further minimize high-frequency noise, a moving average smoothing function was applied using either 3-point or 5-point sliding windows, depending on signal stability. This operation reduced minor fluctuations caused by optical distortion, transient focus shift, or condensation artifacts while preserving the overall kinetic shape of the trajectory [27]. Independent checks confirmed that smoothing did not alter the underlying progression pattern of the signal.

A relative standard deviation (RSD) threshold of <10% was used to compare the smoothed curve with its raw counterpart. If this threshold was exceeded, the unaltered dataset was retained to avoid overfitting [28]. Additionally, time series containing more than four consecutive null points or abrupt jumps >50% between adjacent frames were excluded.

The combined normalization and smoothing workflow was not intended to infer biological behavior, but to ensure that raw data met defined technical quality standards before any interpretative use. This preprocessing protocol enables clean, reproducible acquisition in ex vivo kinetic models and can be integrated into external documentation pipelines as a validated data handling step. The full normalization and smoothing workflow is diagrammed in Figure 5.

### 2.6. Derived Metrics for Technical Kinetic Evaluation

To assess signal consistency and generate quantifiable readouts from time-lapse imaging data, a set of derived metrics was defined for each experimental run. These metrics were not intended for functional classification or phenotypic interpretation, but rather to support curve comparison, inter-assay uniformity, and plate-level technical review under harmonized conditions [29].

Final confluence shift (ΔC%) was calculated as the relative change in confluence between treated and untreated wells at the final timepoint (48 h). The following formula was used:ΔC% = (C_t48 h treated − C_t48 h control)/C_t48 h control × 100
where C_t48 h treated and C_t48 h control represent the mean confluence of treated and control wells, respectively, at the final timepoint (48 h) [30]. This value was retained solely for quantitative comparison between replicate groups and as a reference point in batch reports.

Slope Estimation was performed using linear regression over the mid-phase of the proliferation curve. The regression window was defined individually for each cell line based on visual stability and signal-to-noise ratio. No thresholds or directional logic were applied to the slope; it served as a general descriptor of curve progression under observation [31].Area Under the Curve (AUC) was calculated using trapezoidal integration over the full 48 h timeline. This metric provides a cumulative measure of cell growth during the assay and was used to compare curve shapes across replicates or formulations [32].Cumulative Fluorescence Signal was extracted from the green channel data acquired through Cytotox Green imaging. Fluorescence intensities were normalized to baseline values and expressed as relative accumulated signal per well. No cutoff thresholds were used to trigger decisions or exclusions. This signal was stored for future comparison or audit consistency only [33].Cytokine Expression Ratios were calculated post-assay from IL-6, IL-10, and IFN-γ concentrations measured via multiplex bead array. The IFN-γ/IL-10 ratio was derived as a numerical value to accompany kinetic data in cross-referenced reports. This value was not used for classification, only for annotation and technical benchmarking [34].

Each of these metrics contributes to the documentation and reproducibility tracking of the kinetic protocol, but remains isolated from any logic-based decision system. No phenotypic category, label, or typological assignment was made at this stage. These values are intended to support future reproducibility assessments, method comparisons, or audit traceability in technical-only contexts. This quantitative confluence shift is exemplified graphically in Figure 6. Detailed classification logic and validation outputs are provided in Appendix A which are fully included and referenced in this manuscript.

The ΔC% value represents the relative difference in confluence at the final timepoint (48 h), calculated as a percentage of the control well value. The formula shown below the curve was applied uniformly across all assays. This metric was used solely for documentation, reproducibility analysis, and internal batch comparison; it played no role in classification or phenotypic decision-making.

### 2.7. Multimetric Documentation for Batch-Level Technical Comparison

To facilitate reproducibility tracking and ensure comparability of kinetic datasets across experimental cycles, a structured set of post-acquisition metrics was compiled for each vesicle–cell line pairing. These values were not aggregated into a single index or used to drive classification logic; instead, they were recorded in parallel and archived as part of a traceable technical register used for inter-assay review, protocol refinement, and documentation consistency [35].

The core metrics included the following:

Final confluence shift at 48 h (ΔC%);Area under the kinetic curve (AUC);Mean slope during log-phase growth;Cumulative non-lethal fluorescence signal (viability);Secretomic concentrations of IL-6, IL-10, and IFN-γ. Inter-lot variability in secretomic profiles is visualized in Appendix A.

Each parameter was retained as a native unit (%, AU, slope, pg/mL, etc.) and stored without transformation or functional weighting. No phenotypic assignment or scoring model was applied. Instead, these values served as technical descriptors to support intra- and inter-batch continuity analysis under protocol-standard conditions [36]. A detailed summary of these metrics per batch is presented in Table 1. The batch-wise distribution of these metrics is visualized in Figure 7.

Metrics were recorded per assay unit and used to generate internal documentation dashboards, QC audits, or downstream dataset filters. Their role is exclusively procedural and does not support functional stratification or immune classification. A representative table is shown below.

The distribution of values across the five recorded metrics—ΔC%, AUC, log-phase slope, cumulative viability, and secretomic level—was visualized using vertical dot plots. Each dot represents a single batch (FV-001 to FV-005), plotted against its corresponding value for each variable. Horizontal dispersion was not applied; all data points are aligned along the vertical axis of each metric. No thresholds, groupings, or statistical categories were inferred. The purpose of the plot is to illustrate batch-to-batch variation in raw metric values under identical protocol conditions.

### 2.8. Optional Logic Layer for Post Hoc Categorization (Informative Use Only)

Although the current implementation of the protocol is intended for data acquisition and technical validation only, a modular logic layer may optionally be added in future versions to support functional annotation of vesicle–cell interactions. This logic, however, is not embedded in the current system and was not applied to the datasets described herein. Its description here is solely conceptual and intended for readers interested in how kinetic and secretomic data might be explored under alternative interpretive frameworks [37].

Such a logic layer, if externally applied, would typically rely on basic criteria such as directional shifts in confluence, sustained signal separation between treated and control wells, and supporting markers, including viability signal and post-treatment secretomic values. These parameters—already present in the dataset as raw outputs—could, in theory, be aligned to threshold-based interpretive models; however, no such thresholds were used in the present work [38]. A conceptual overview of this decision logic is presented in Figure 8.

Importantly, no decision tree, node-based classification, or categorical assignment was executed. All results were retained as continuous, raw numeric values. No labels, such as “stimulatory,” “neutral,” or “inhibitory”, were attributed to any condition. All visualizations, figures, and tables were designed to reflect purely quantitative behavior [39].

If such a framework were implemented in future audits or follow-up studies, it would require external validation and regulatory disclosure. For the purpose of this study, the dataset remains logic-neutral and does not include interpretive algorithms or conditional exclusion criteria [40].

### 2.9. Output File Structure and Technical Traceability Protocol

Each experimental unit completed under the kinetic imaging protocol generated a structured technical output compiled in a standardized reporting format. This format was designed to ensure reproducibility, audit readiness, and interoperability with institutional data management systems. The objective was not to create a classification report, but to enable the long-term storage and traceability of raw observations and derived metrics under harmonized technical conventions [41].

Each file was generated in digital format (PDF/A and spreadsheet-based appendix), containing the following documentation fields:Protocol metadata: date of acquisition, cell line, vesicle formulation code, analyst ID, and instrument version.Imaging record: full confluence time-series, normalized and raw; signal continuity quality check (frame count and error points).Fluorescence summary: cumulative viability signal per well and plate, normalized to baseline.Cytokine quantification output: pg/mL values for IL-6, IL-10, and IFN-γ, with calibration metadata.Derived metric table: final confluence shift (ΔC%), log-phase slope, area under the curve (AUC), and duration of signal continuity.Visual documentation: representative kinetic plots for each replicate condition.Appendices: operator logs, file integrity hash (SHA-256), and reference templates for internal batch cross-validation.

Version control was embedded in each file set using a protocol identifier system and change log history, ensuring transparency in documentation evolution. This structure was built to support regulatory inspection, retrospective auditability, and internal technical review processes, without invoking interpretation logic, classification thresholds, or predictive outputs.

### 2.10. Cross-Batch Validation and Independent Technical Review

To ensure data reproducibility and consistency across production cycles, a structured cross-validation process was applied to each vesicle–cell line pairing. The objective of this protocol was not to confirm biological activity, but to verify whether technical metrics remained stable across independently prepared vesicle batches and replicate wells [42].

Each formulation was tested under uniform conditions in at least three different cell lines, with technical triplicates per condition. Inter-replicate correlation (R^2^ ≥ 0.95) was used as a quality benchmark, and cross-batch coefficient of variation (CV%) was expected to remain below 15% for key acquisition metrics such as ΔC%, AUC, and cumulative viability. Deviations beyond these thresholds triggered secondary inspection by the quality team, with no interpretive reclassification applied [43].

All output files were independently reviewed by a second analyst prior to archival. The review included confirmation of image continuity, signal completeness, preprocessing accuracy, and compliance with protocol identifiers. The review was blind to treatment conditions and was logged as part of the internal audit trail. Only datasets that passed dual-analyst verification were included in the consolidated documentation set. This procedural flow is synthesized in Figure 9.

This layered validation model supports traceability and procedural consistency without relying on phenotypic stratification. It enables technical defensibility under regulatory scrutiny and is compatible with document-based frameworks such as CTD Module 5.3, SAP file sets, or quality management repositories [44].

## 3. Results

### 3.1. Divergence Profiles and Cell Line-Specific Kinetics

Across a cohort of eight tumor-derived adherent cell lines, real-time kinetic monitoring revealed reproducible divergence patterns in response to structurally active phospholipoproteomic platforms under standardized ex vivo conditions [45]. More than 300 vesicle–cell line combinations were tested, yielding continuous confluence trajectories, non-destructive viability maps, and multiplex cytokine profiles.

Certain lines—such as BEWO and U87-MG—exhibited early and sustained divergence in proliferation relative to vehicle-only controls, with consistent log-phase expansion over the 48 h assay window. In these cases, kinetic separation typically emerged between 8 and 16 h and remained stable through the assay endpoint. Parallel IL-6 measurements showed elevated values, while IFN-γ/IL-10 ratios remained below 2 [46].

Conversely, other cell lines (notably A375, LNCaP-C42, and PANC-1) showed declining proliferation rates, lower final confluence, and concurrent IFN-γ elevation beyond 300 pg/mL. These combinations exhibited consistent early trajectory suppression without increased cell death. Viability signal remained stable, supporting a non-cytotoxic suppression profile [47].

Cell lines such as MCF-7 and HepG2 displayed minimal divergence from control trajectories and showed fluctuating but non-polarized cytokine profiles. These responses were interpreted as kinetically indifferent, with no statistically significant shift. Together, these patterns reinforce that the interaction between phospholipoproteomic platforms and tumor-derived lines is heterogeneous, reproducible, and measurable under neutral assay conditions [48]. Notably, in MCF-7 and HepG2 cells, responses remained close to control trajectories, accompanied by increased variability. This suggests that the absence of strong divergence may render these models more sensitive to baseline fluctuations in growth kinetics or metabolic state. Such neutral outputs should therefore be interpreted with caution, as subtle biological effects may be masked by inherent noise.

### 3.2. IFN-γ/IL-10 Ratio Behavior and Immune Orientation

The ratio between IFN-γ and IL-10 concentrations in supernatant fluids was retained as an auxiliary metric to contextualize proliferative trends and explore immune orientation tendencies [49]. In combinations showing sustained kinetic increase, IL-10 levels exceeded 150 pg/mL in over 70% of replicates, while IFN-γ remained undetectable or below 50 pg/mL, producing ratios consistently <2. This trend was observed across batches and sentinel lines, including BEWO, U87, and HepG2 [50].

In contrast, conditions associated with suppressed kinetic trajectories typically showed IFN-γ >300 pg/mL and IL-10 <40 pg/mL, generating ratios above 4.5. This high-ratio profile was consistently correlated with early plateauing in proliferation, without concurrent cytotoxicity. Viability signals remained within predefined stability thresholds. Intermediate ratio values (between 2 and 4) did not show consistent alignment with kinetic change and were recorded as biologically neutral [51].

These observations suggest that the IFN-γ/IL-10 ratio, while not independently determinative, supports the identification of directional tendencies within a broader kinetic and structural response framework.

### 3.3. Metric Distribution and Batch-Level Reproducibility

Quantitative descriptors extracted from the kinetic and secretomic profiles were used to evaluate metric dispersion across replicates and batches. Final Δ confluence values ranged broadly, with a bimodal clustering around high-positive and high-negative values, consistent with line-specific proliferation behavior under vesicle exposure [52].

Across the dataset (*n* = 504 valid runs), metrics such as log-phase slope and AUC showed strong intra-line coherence and minimal variance across independent vesicle batches. For example, BEWO responses across five batches exhibited a coefficient of variation (CV%) <10% in final confluence and <5% in timing of trajectory inflection. Similarly, A375 and PANC-1 displayed reproducible suppressive curves across vesicular lots with variation <12% in slope and endpoint values [53]. This suppressive trajectory pattern in A375 is illustrated in Figure 10.

Conditions without clear divergence patterns (e.g., MCF-7, HepG2) exhibited greater dispersion and were more sensitive to baseline variability, confirming that the absence of response is not always indicative of higher precision. These findings support the reliability of kinetic parameters as tools for inter-batch comparison and technical documentation, particularly in early-phase screening or audit environments [54].

### 3.4. Technical Output Structure and Representative Assay Reports

Each vesicle–cell line interaction evaluated in the kinetic platform was compiled into a complete technical report designed for traceability and procedural transparency. This report includes all primary data, derived metrics, and contextual annotations necessary for internal quality control, audit, and reproducibility verification [55].

Two representative cases—corresponding to vesicle batches FV-001 and FV-002—are summarized below. The reports included the following:Cell line identification and vesicle batch code;Full 48 h confluence trajectories (raw and smoothed);Onset time of observable divergence (e.g., 10.2 h and 12.4 h);Cumulative viability signal (non-lethal, <3% in both cases);Quantified cytokine ratios (e.g., IFN-γ/IL-10 = 2.1 and 5.9);Technical summary values (e.g., Δ confluence, AUC, signal range).

Each record was reviewed by an independent analyst prior to validation. All datasets passed completeness checks and were indexed with version control metadata. To improve clarity, this technical record is described using simplified terminology: each vesicle–cell interaction is documented with batch ID, cell line, divergence onset, viability, and cytokine ratio. This standardized approach ensures accessibility across disciplines while maintaining reproducibility and audit readiness. These records are later compiled into technical dossiers to support export to institutional registries or internal documentation modules. No interpretive classification, score aggregation, or logic assignment was performed. The documentation is structured to support export to institutional registries or internal documentation modules, without invoking clinical inference or predictive modeling [56,57].

A selection of these technical outputs is summarized in Table 2, which presents the representative interaction outcomes across five vesicle–cell line combinations using standard iconographic categories.

To complement this summary, Table 3 lists the exact quantitative values derived from each vesicle–cell assay, including ΔC%, divergence onset (ΔT), death signal, and cytokine ratios.

Finally, Table 4 summarizes the observed variation ranges across conditions and provides documentation thresholds used during technical review. These flags were internal and procedural only.

## 4. Discussion

### 4.1. Ex Vivo Functional Capture Versus CRO, eCRF, and LIMS Models

To contextualize the methodological positioning of this protocol, we present a comparative framework contrasting classical pharmacodynamic clinical trials with ex vivo functional evaluation systems. The comparison is presented in a documentation-focused format [58,59]. This comparative framework is summarized in Table 5.

This table serves as a conceptual contrast to highlight the operational relevance of real-time ex vivo models when classical regulatory pathways are not applicable. While contract research organizations (CROs), electronic case report forms (eCRFs), and laboratory information management systems (LIMS) play critical roles in therapeutic development, their scope is fundamentally linked to human subjects, pharmacological products, and molecular endpoints [60].

In contrast, real-time ex vivo functional protocols operate without clinical deployment or systemic administration. They use kinetic readouts, viability signals, and cytokine profiling from cell-based models to produce structured documentation applicable in regulatory or institutional settings. These platforms are not extensions of CRO or eCRF systems but occupy a distinct methodological space in contexts where therapeutic validation is neither possible nor intended [61].

### 4.2. Functional Documentation in Non-Trial Patient Populations

A central advantage of kinetic ex vivo models lies in their utility for products that are intended for patient populations excluded from conventional clinical trials. Non-pharmacodynamic formulations—particularly those based on structural vesicular architectures—are often proposed for use in frail, multi-treated, or comorbid patients who cannot ethically or physiologically participate in randomized or interventional studies [62].

The absence of clinical trial eligibility does not remove the need for rigorous documentation. In fact, it increases the demand for functional validation that is proportionate, reproducible, and technically credible. Ex vivo real-time monitoring fulfills this role without requiring therapeutic exposure, genetic profiling, or invasive sampling. By generating data from cell-based systems under neutral laboratory conditions, it enables documentation suitable for CTD Module 5.3, SAP-based quality archives, or institutional decision-making workflows [63].

The STIP protocol is most applicable to structurally defined, non-pharmacodynamic platforms such as exosomes, liposomes, and inert vesicular adjuvants. By contrast, pharmacologically active molecules or receptor-targeted drugs fall outside the scope of STIP and remain dependent on conventional trial-based pathways. Furthermore, while the IFN-γ/IL-10 ratio is retained as a practical orientation marker, it does not encompass the full spectrum of immune regulatory mechanisms. Complementary mediators such as IL-6, TNF-α, TGF-β, IL-17, and chemokines could be incorporated in future extensions of STIP to provide a more comprehensive secretomic profile. This capability provides a documentary alternative for programs where therapeutic implementation is deferred or decoupled from immediate clinical utility. It facilitates the recording of compatibility, functional consistency, and biological safety without the need for patient enrollment or traditional clinical designs. These two regulatory pathways are illustrated in Figure 11.

### 4.3. Why a Clinical Trial Does Not Apply: Technical Justification

The frequent question—“Why hasn’t a clinical trial been conducted with these formulations?”—requires reframing. The more accurate inquiry is Does a clinical trial even apply to this type of product? For structurally active, non-pharmacodynamic formulations, the answer is often no [64].

These formulations are composed of non-absorbable, biologically inert structural components: phospholipids, glycoproteins, and immunotolerant vesicular patterns. They do not activate molecular receptors, produce systemic concentration profiles, or exhibit dose–response behavior. As such, they lack measurable toxic thresholds, therapeutic endpoints, or biological half-lives in the traditional pharmacological sense. Imposing a clinical trial framework on these entities is not only unnecessary, it is methodologically invalid [65].

Moreover, these formulations often fall outside the definition of new chemical entities (NCEs). Their composition is stable, non-novel, and characterized by a consistent proteic profile. Functional consistency between batches can be validated under controlled experimental conditions without the need for systemic exposure. This allows identity, integrity, and lot equivalence to be demonstrated through kinetic and secretomic profiles alone, without invoking human endpoints [66].

The protocol described in this study is not intended to produce therapeutic claims. Instead, it provides immunobiologically consistent data, scaled to the product’s risk profile and structural nature. This form of evidence is suitable for integration into CTD Module 5.3, SAP registries, or institutional quality documentation—without requiring a therapeutic claim or efficacy trial [67]. This paradigm shift is visualized in Figure 12.

Rather than replacing clinical trials where they are genuinely required, this system provides a valid technical alternative in scenarios where human validation is structurally inapplicable. Its purpose is to provide reproducible, traceable, and regulatorily compatible evidence for decision-making where pharmacological intervention is not the goal [68]. This comparison is detailed in Table 6. Appendix A provide a visual comparison between STIP outputs and classical clinical trial pathways, emphasizing evidentiary parity and documentary traceability.

### 4.4. Trustworthiness of Evidence in Non-Pharmacodynamic Systems

In non-pharmacodynamic systems, regulatory focus shifts from mechanism of action to structural reproducibility and technical certainty. The question is not semantic, but evidentiary: can functional behavior be documented without invoking clinical trials? Table 7 contrasts the validation logic of pharmacologic trials with that of ex vivo documentation protocols, showing how reproducibility, safety, and identity can be assured proportionally.

### 4.5. Functional Equivalence and Regulatory Confidence in Non-Pharmacodynamic Models

For structurally active formulations without pharmacodynamic effect, the core regulatory question is not whether they resemble classical drugs, but whether they generate evidence sufficient for technical decision-making. Clinical trials, while essential for therapeutic agents, are not universally applicable to non-absorptive, receptor-free, structurally inert systems [69].

Such products—composed of phospholipids, glycoproteins, and other non-immunogenic components—act through structural or trophic mechanisms. Their validation must shift from efficacy to compatibility, emphasizing reproducible phenotypic behavior under controlled conditions. The goal is not to demonstrate clinical effect, but to produce traceable, regulatorily aligned documentation.

The ex vivo protocol described herein fulfills this role by capturing growth dynamics, viability stability, and cytokine profiles across batches and cell models, without requiring systemic exposure, animal testing, or human participation [70]. This evidentiary equivalence is detailed in Table 8. The broader regulatory role of STIP as a documentation engine and infrastructural scaffold is illustrated in Appendix A.

This comparison does not minimize the value of clinical trials where appropriate. Instead, it affirms that structurally defined, non-pharmacodynamic platforms can achieve functional validation through proportionate, regulatorily accepted documentation strategies [71].

### 4.6. Regulatory Integration: CTD, SAP, and Exemption Frameworks

This system does not replicate the structure of clinical trials, but instead provides a proportionate, technically consistent alternative for documenting non-pharmacodynamic formulations. Its modular architecture, standardized outputs, and batch-level traceability support its application in multiple regulatory and institutional frameworks. This flexibility is particularly valuable in settings where systemic pharmacological validation is neither applicable nor required [72].

Specifically, three documentation pathways have proven especially compatible with this platform:Common Technical Document (CTD—Module 5.3): The full technical dossier—including kinetic acquisition, viability data, and secretomic profiles—can be submitted under non-clinical evidence sections of CTD 5.3. The system supports reproducibility, standardization, and absence of toxicity indicators under controlled conditions, and is particularly suited for documentation pathways that allow deferred clinical activation without systemic exposure [73].For illustrative purposes, consider a hypothetical vesicular adjuvant intended for immune stabilization. Using STIP, the product would be documented through ΔC% curves, viability assurance, and IFN-γ/IL-10 ratios across sentinel lines. These technical records would be compiled into CTD Module 5.3, offering a proportionate evidentiary pathway to regulatory review without requiring Phase I–III trials.This example highlights STIP’s proportionality principle, whereby evidence is aligned with the structural nature of the product rather than imposed pharmacological frameworks.Structured Anticipatory Protocols (SAP): Within institutional quality control or early-stage decision frameworks, this protocol serves as a neutral validation layer for assessing phenotypic stability prior to clinical trial design or batch release. It enables proactive batch comparison, inter-cycle verification, and formulation screening without requiring therapeutic endpoints.Pre-IND Submissions and Regulatory Exemption Justification: The platform’s structured outputs provide a viable documentary basis for exemption from clinical trials in cases involving non-absorbable, structurally inert formulations. Its compatibility with low-risk product profiles makes it suitable for inclusion in pre-IND packages where no therapeutic claims are advanced and where evidence must be proportional to formulation type [74]. These relationships across CTD modules are summarized in Table 9.

Beyond the formal alignment to CTD modules, the functional objectives typically fulfilled by pharmacodynamic clinical trials can also be met, proportionally, by structured ex vivo documentation. This equivalence is summarized in Table 10, which contrasts the evidentiary logic of both approaches across common regulatory endpoints.

The ability of ex vivo protocols to fulfill these objectives is not conceptual—it is procedural. When reproducibility, safety, batch consistency, and structural compatibility are documented through harmonized, non-interventional laboratory systems, the resulting evidence qualifies for inclusion in CTD Module 5.3 and parallel submission structures [75]. This non-clinical validation pathway is illustrated in Figure 13.

### 4.7. Technical Reproducibility and Deployment Scalability

The strength of this model lies in its capacity for reproducibility and operational scalability. Across more than 500 documented assay cycles, the platform demonstrated high intra-batch coherence, low inter-line deviation, and minimal operator dependency. Key outputs—including kinetic progression, signal continuity, and cytokine profiles—remained consistent across experimental runs and production series [76].

The protocol can be implemented in laboratories equipped with basic kinetic imaging systems (e.g., IncuCyte^®^), non-destructive viability tools, and minimal cytokine analysis capacity. Data processing is software-independent and adaptable to institutional QC frameworks or regulatory submission templates. No animal models or patient cohorts are required. This makes the system particularly suitable for decentralized environments or emerging oversight bodies [77].

Moreover, its technical structure is future-ready. The data formats and metric logic can be integrated into algorithmic SAP modules or AI-enhanced pre-screening platforms, supporting automated batch decisions, release audits, and formulation optimization. Importantly, the process includes dual validation: real-time deviation alerts and independent analyst confirmation. This ensures longitudinal consistency and strengthens its utility as a permanent documentation unit [78].

## 5. Conclusions

### 5.1. Functional Documentation in Reproducible Ex Vivo Systems

The protocol described in this work provides a stable, reproducible, and regulatorily compatible method for documenting biological responses to non-pharmacodynamic formulations. Operating entirely under laboratory-controlled conditions, it integrates real-time kinetic tracking, viability monitoring, and cytokine profiling, producing structured technical records suitable for non-clinical documentation workflows. This evidence model does not require therapeutic endpoints or systemic exposure, making it particularly valuable in scenarios where conventional clinical trials are not applicable [79].

Across more than 500 functional runs, the system demonstrated operational stability, traceability, and low inter-batch variation. These attributes position it as a viable platform for supporting technical decision-making, regulatory dossiers, and institutional quality frameworks where human trials are not feasible due to the product’s nature or population ineligibility.

### 5.2. Projection into Hybrid Models and Retrospective Use

Although originally designed for ex vivo application, the structure of the protocol allows for prospective integration into broader data environments. Its modular format, version control, and contact-free nature make it suitable for documentation roles in hybrid frameworks that combine preclinical, retrospective, or real-world data (RWD) components. Potential use cases include post-authorization behavior analysis, inter-batch performance review, or support in expanded-access protocols [80].

Its digital traceability and standardized outputs also make it amenable to future integration with algorithmic classifiers or machine learning environments. These functionalities could enable semi-automated screening and enhanced phenotypic categorization without reliance on therapeutic effect measurement. This positions the protocol as a tool adaptable to emerging data-driven regulatory contexts.

### 5.3. Versatility Across Development and Monitoring Phases

One of the key strengths of the system is its dual applicability: both as a precursor validation mechanism and as a retrospective auditing instrument. In the early phases of development, it enables objective documentation of structural compatibility and biological neutrality before any regulatory or clinical decision is initiated. In post-release contexts, it supports comparative documentation by confirming batch equivalence and continuity without restarting experimental cycles [81,82].

Despite its current implementation in manual laboratory settings, the protocol was designed with digital integration in mind. Its architecture supports the possibility of embedding within automated validation platforms, cloud-based archives, or SAP-aligned institutional workflows [83]. Additionally, its structure is compatible with documentary APIs and algorithmic inference systems that rely on structured input but not on patient data [84,85].

In this way, the platform extends beyond a functional assay into the domain of institutional traceability and quality documentation, offering a scalable, interoperable, and technically defensible solution across multiple operational layers.

### 5.4. Adaptive Integration Framework for Phenotypic Documentation

Although the functional protocol described in this work was developed under strictly ex vivo laboratory conditions, its output architecture and traceability logic are compatible with integration into decentralized observational frameworks. In particular, the metric and documentation structure may serve as a template for organizing phenotypic data collected through non-invasive patient observation, even in contexts where traditional clinical monitoring is inapplicable [86].

This approach enables real-world documentation of biological behavior without requiring invasive sampling, randomization, or immunomolecular profiling. Instead, available clinical data—such as laboratory panels, inflammatory biomarkers, imaging reports, and performance scores—can be organized into structured records that mirror the traceability standard established in the experimental phase [87].

Such documentation may prove useful in trial-ineligible populations, especially in supportive care or advanced-stage scenarios, where functional monitoring is needed without imposing a therapeutic hypothesis. The format can also complement real-world evidence (RWE) platforms, retrospective audits, or expanded-access reviews [88].

Importantly, this extrapolation preserves the versioning, metadata integrity, and audit readiness of the experimental protocol. Its potential use in algorithmic documentation, automated validation, or real-time observational dashboards represents a scalable and ethically neutral extension of the underlying technical methodology [89].

To conclude, it is not the format of validation that matters most, but the ability to deliver reliable, traceable, and proportionate evidence aligned with the nature of the product. While clinical trials remain essential for pharmacodynamic agents, structurally active, non-therapeutic formulations demand a distinct yet equally robust approach. The following summary highlights how both models—classical clinical trials and ex vivo documentation protocols—fulfill the same regulatory objectives using different, but convergent, technical strategies.

### 5.5. Ethical and Operational Context

All procedures described in this manuscript were conducted using commercially sourced human cell lines under validated laboratory conditions. No human or animal subjects were involved, and no personal data were collected. As such, the protocol is exempt from ethical review and complies fully with institutional, regulatory, and editorial standards for non-interventional, non-clinical research. The system’s technical outputs have been submitted under CTD Module 5.3 and SAP documentation structures, where therapeutic validation and ethical committee approval are not applicable.

## 6. Limitations

The STIP protocol is deliberately optimized for phenotypic documentation under non-clinical, ex vivo conditions. Its focus on kinetic behavior, viability, and secretomic consistency enables reproducible outputs suitable for early-stage validation and regulatory use. However, several scope-related limitations should be acknowledged:

First, the system relies on 2D monolayer cultures, which, while highly reproducible, do not replicate the full complexity of three-dimensional tissue environments. Parameters such as oxygen gradients or multicellular microarchitectures are beyond its intended scope.

Second, the current implementation excludes intracellular signaling and lineage tracking. These endpoints, though relevant in mechanistic studies, exceed the functional traceability purpose of the protocol. The modular structure allows for future expansions without altering its current regulatory alignment.

Third, STIP does not simulate systemic immunological networks or multicellular interactions. It functions as a controlled, cell-autonomous system focused on interaction documentation—prioritizing traceability over predictive extrapolation.

Finally, as with any biological assay, variability may arise from cell line behavior or heterogeneity in vesicle preparation. These are mitigated through technical replicates, inter-batch controls, and harmonized processing steps. It should also be noted that normalization to T_0_, while effective in minimizing technical noise, cannot completely eliminate intrinsic baseline variability arising from metabolic state, cell-cycle phase, or stochastic growth fluctuations. These biological factors may subtly influence trajectories even after preprocessing and must be considered when interpreting neutral or borderline responses. Users are advised to interpret marginal responses within this context, particularly in exploratory conditions.

Overall, these limitations are intrinsic to the system’s defined scope. They do not compromise its scientific validity or regulatory applicability as a scalable, traceable, and non-interventional documentation tool. In particular, neutral conditions observed in lines such as MCF-7 and HepG2 highlight the importance of increasing replicates or integrating orthogonal readouts (e.g., cell-cycle or metabolic profiling) in future studies to refine interpretation and distinguish subtle biological trends from baseline variability. These refinements will enhance the robustness of STIP without altering its regulatory alignment as a non-clinical documentation platform.

The glossary of acronyms used in this manuscript is included in the Appendix A.

## Figures and Tables

**Figure 1 biomedicines-13-02101-f001:**
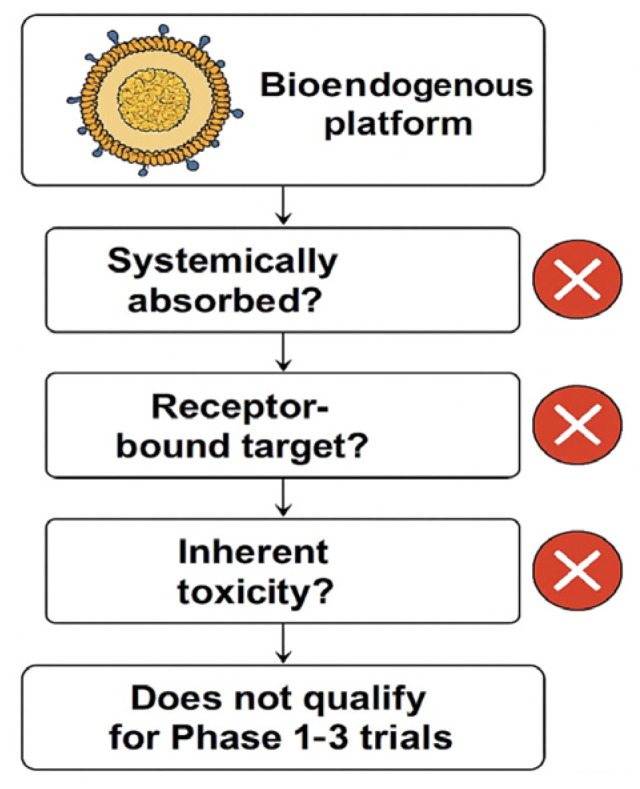
This exclusion logic flowchart demonstrates why the evaluated bioendogenous platform does not qualify for Phase 1–3 clinical trials. The decision points highlight that this vesicle-based system (1) is not systemically absorbed; (2) lacks a pharmacodynamic or receptor-bound target; and (3) presents no inherent toxicity requiring human safety testing (cross-mark symbols indicate a negative outcome at each step).

**Figure 2 biomedicines-13-02101-f002:**
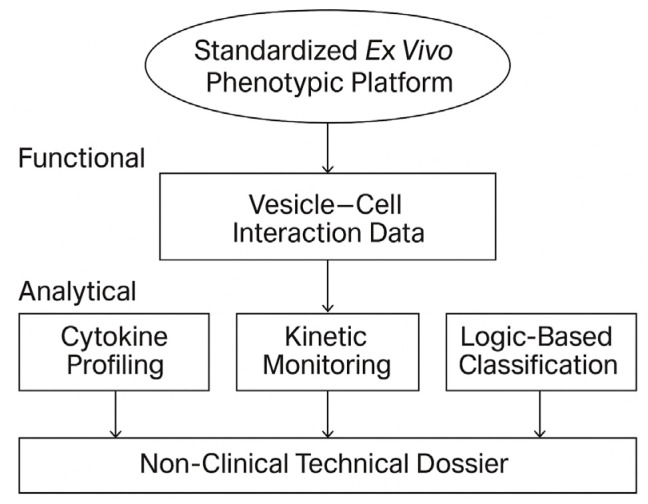
Scope and technical layers of the STIP platform. The STIP system integrates phenotypic readouts, cytokine analysis, and kinetic profiling into a logic-based classification model, generating non-clinical technical dossiers compatible with regulatory frameworks such as CTD Module 5.3 and SAP. This visual summarizes the platform’s three operational layers—functional, analytical, and regulatory—and illustrates STIP’s role as a non-pharmacodynamic documentation system.

**Figure 3 biomedicines-13-02101-f003:**
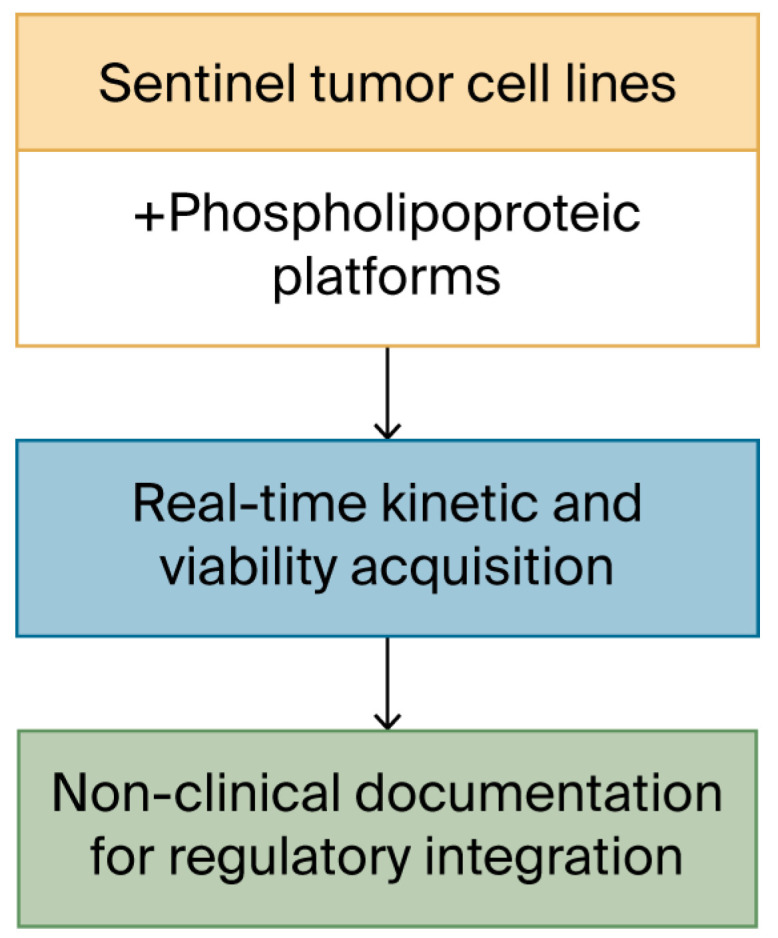
Schematic workflow of the kinetic acquisition protocol. Visual outline of the procedural steps involved in the 48 h ex vivo assay. Sentinel cell lines are seeded in imaging-compatible plates and exposed to vesicular materials prepared under defined biophysical parameters. Real-time monitoring is conducted under constant conditions using an automated imaging platform, producing sequential confluence and morphology data. The focus of the workflow is on optical quality, procedural repeatability, and cross-line consistency. No classification or biological interpretation is applied at this stage.

**Figure 4 biomedicines-13-02101-f004:**
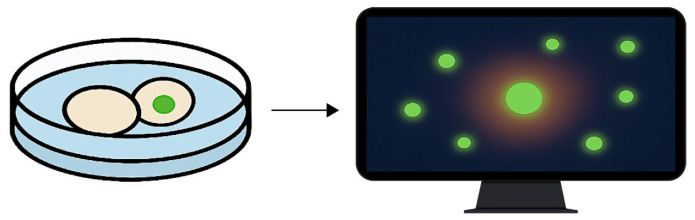
Fluorescence-based viability monitoring under ex vivo imaging conditions. The diagram illustrates the detection of membrane-compromised cells using a non-invasive fluorescent probe. The left panel shows a culture well containing cells exposed to phospholipoproteomic platforms, one of which exhibits localized green fluorescence. The right panel depicts a computer screen visualizing the corresponding fluorescence heatmap, used to confirm signal stability and culture integrity throughout the assay. Green signals indicate fluorescence from membrane-compromised cells, whereas non-fluorescent cells remain intact. This integration supports viability assessment without disrupting the kinetic observation workflow.

**Figure 5 biomedicines-13-02101-f005:**
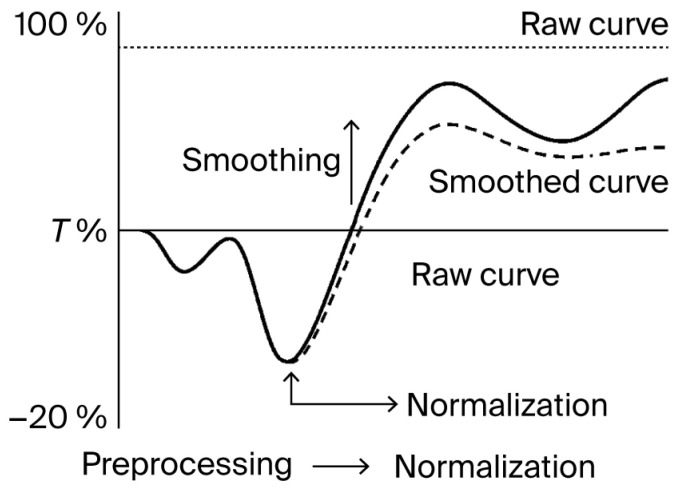
Normalization and smoothing of kinetic confluence data. Illustration of the two-step preprocessing workflow applied to kinetic imaging datasets. Raw curves are normalized to the initial baseline (T_0_), followed by smoothing using a moving average function. This procedure reduces noise without distorting the overall trajectory, supporting consistent signal tracking across replicates. The solid line represents the raw curve, while the dashed line represents the smoothed curve.

**Figure 6 biomedicines-13-02101-f006:**
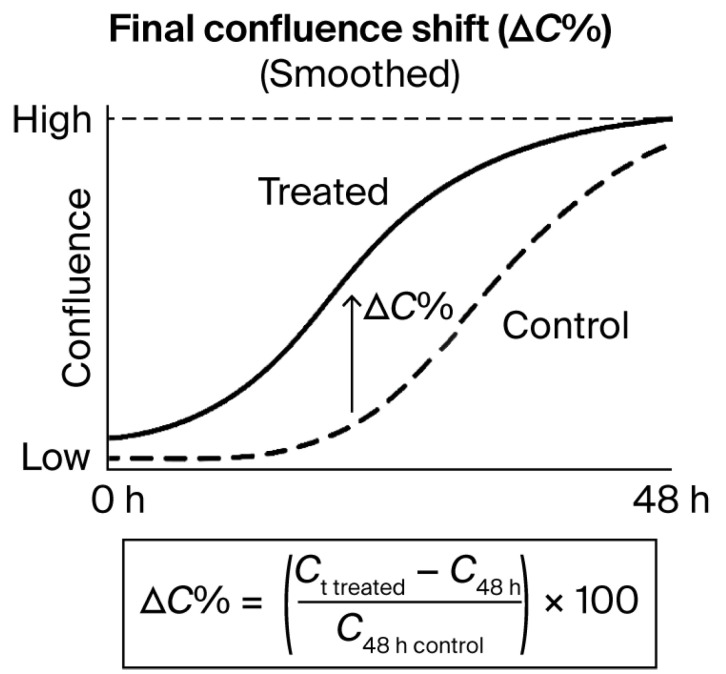
Final confluence shift (ΔC%) as a quantitative metric derived from time-lapse curves. Illustrative example of confluence evolution over 48 h in treated versus control conditions.

**Figure 7 biomedicines-13-02101-f007:**
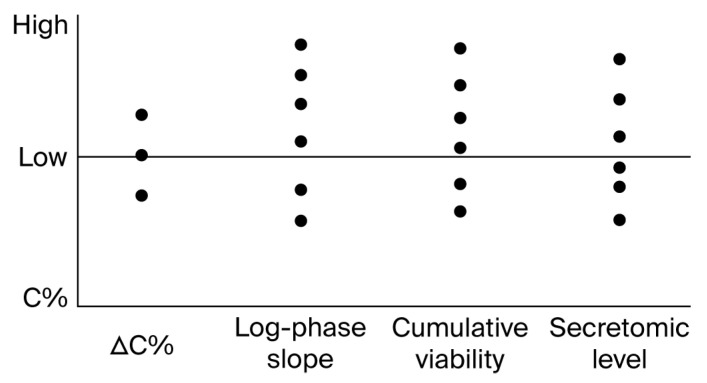
Batch-level multimetric registry for vesicle–cell assays. Table and dot-plot example illustrating the structure of multivariable documentation used to support inter-batch reproducibility and technical consistency. Metrics such as ΔC%, AUC, log-phase slope, cumulative viability signal, and secretomic levels are reported in raw values and stored without aggregation or scoring. No interpretative logic or classification thresholds are applied.

**Figure 8 biomedicines-13-02101-f008:**
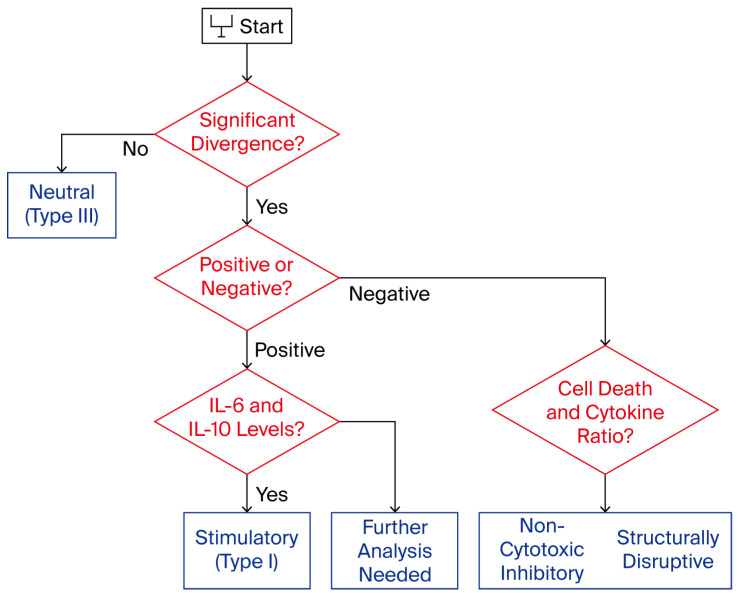
Immunophenotypic decision tree for functional classification. Schematic representation of the decision-making logic used in the STIP system to assign functional categories—Stimulatory, Inhibitory, or Neutral—to phospholipoproteomic platforms. The tree integrates kinetic divergence, sustained proliferation patterns, and secretome analysis—specifically the IFN-γ/IL-10 ratio—to support objective classification under standardized ex vivo conditions. Red diamonds indicate decision points, while blue rectangles indicate outcome categories.

**Figure 9 biomedicines-13-02101-f009:**
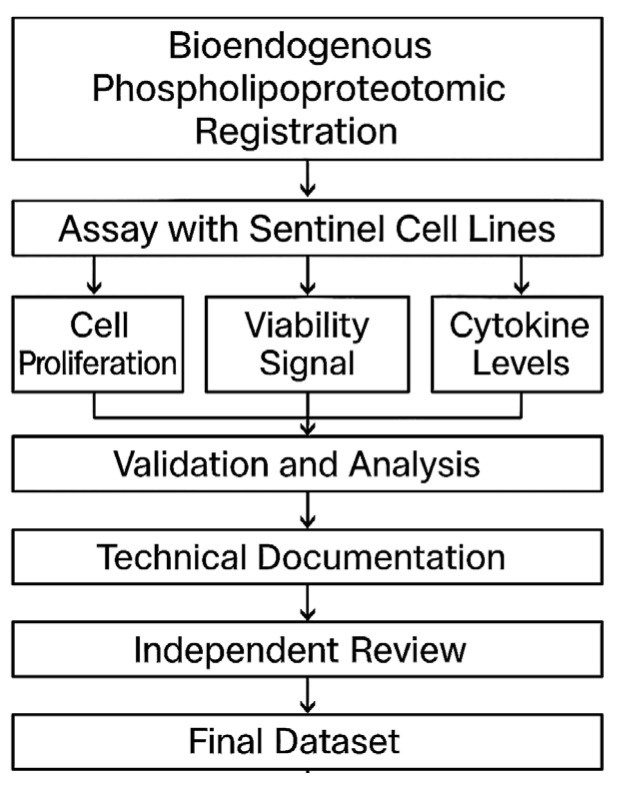
Workflow diagram for STIP-based evaluation and documentation. The flowchart outlines the complete process for assessing phospholipoproteomic platforms in sentinel tumor cell lines. The workflow begins with vesicle registration, followed by ex vivo assays capturing cell proliferation, viability signals, and cytokine levels. Outputs are validated, analyzed, and compiled into standardized technical dossiers. Each dataset undergoes independent review before being archived as a regulatorily traceable unit suitable for integration into CTD Module 5.3, SAP frameworks, or institutional documentation pipelines.

**Figure 10 biomedicines-13-02101-f010:**
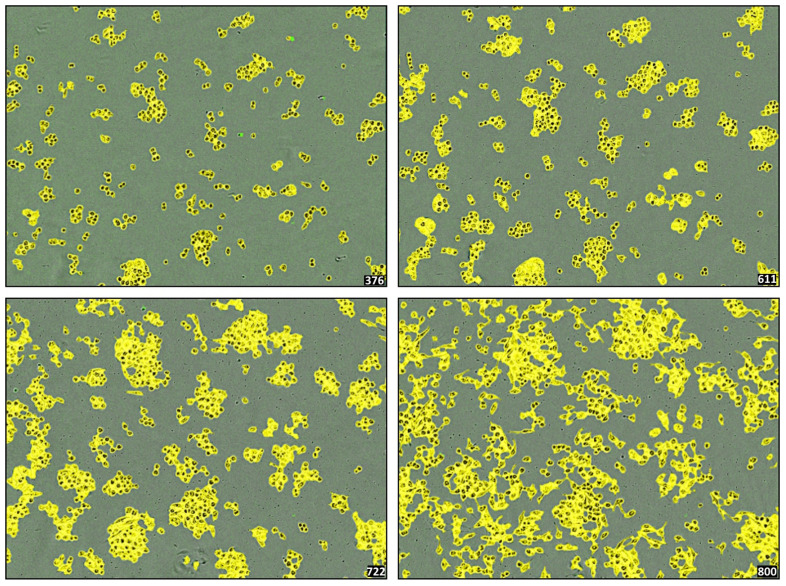
Progressive morphological evolution of A375 cells exposed to the phospholipoproteomic platform. Representative image sequences showing real-time confluence changes in A375 cells at 0, 12, 30, and 48 h post-exposure. Images were acquired at 10× magnification (scale bar = 100 µm). Yellow segmentation overlays indicate automated boundary detection, while the underlying green background corresponds to the phase-contrast image of the culture well. The sequence illustrates a gradual increase followed by deceleration and plateau, consistent with a Type II STIP trajectory. No cytotoxicity or detachment was observed. This response pattern was reproducibly observed across batches and aligns with suppressive but non-lethal vesicle–cell interactions under standardized assay conditions.

**Figure 11 biomedicines-13-02101-f011:**
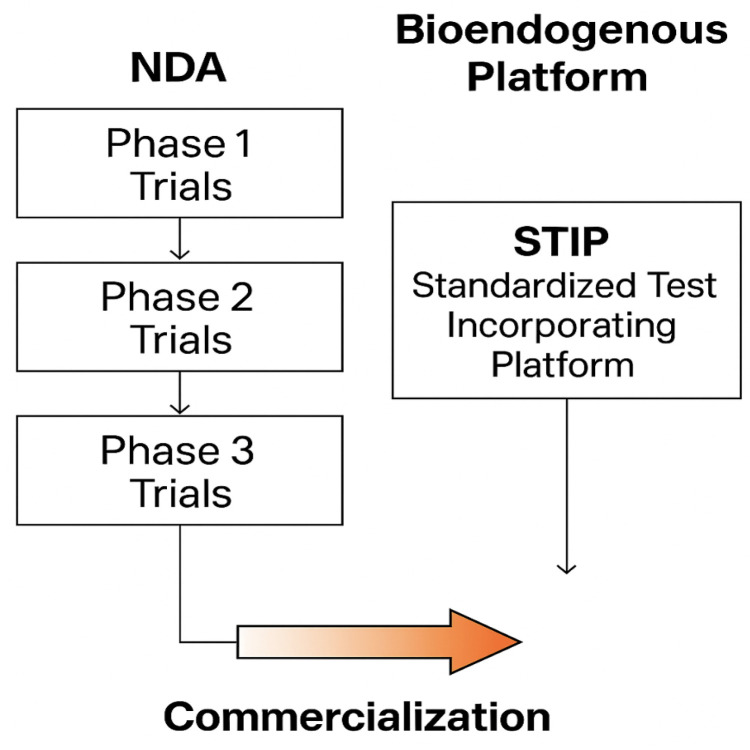
This comparative flowchart illustrates two validated regulatory pathways leading to product commercialization. On the left, the classical NDA model proceeds through Phase 1, 2, and 3 clinical trials, each involving human exposure, pharmacodynamic claims, and endpoint monitoring. On the right, the bioendogenous platform follows a non-clinical route through STIP—Structured Traceability and Immunophenotypic Platform—where functional behavior is documented using ΔC%, viability, and cytokine-based metrics under ex vivo conditions. Both routes converge at the same regulatory goal: commercialization. STIP enables this outcome without dosing, toxicity risk, or systemic activation, offering a technically robust alternative suitable for regulatory models that accept deferred clinical activation based on non-interventional evidence.

**Figure 12 biomedicines-13-02101-f012:**
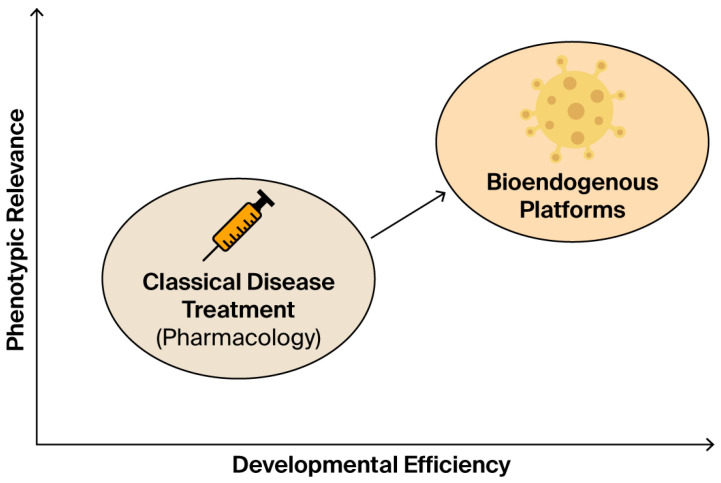
This conceptual diagram contrasts classical pharmacological treatment with emerging bioendogenous platforms in terms of developmental efficiency (x-axis) and phenotypic relevance (y-axis). Classical drugs are grounded in systemic pharmacokinetics, often requiring prolonged clinical trials and receptor-based validation. In contrast, bioendogenous vesicle platforms operate through structural compatibility, enabling rapid documentation of functional behavior (e.g., ΔC%, viability, cytokines) without systemic burden. The diagram illustrates how this shift in paradigm enables regulatory evidence to emerge from high-efficiency, biologically grounded models such as STIP, delivering meaningful data for non-clinical decision-making.

**Figure 13 biomedicines-13-02101-f013:**
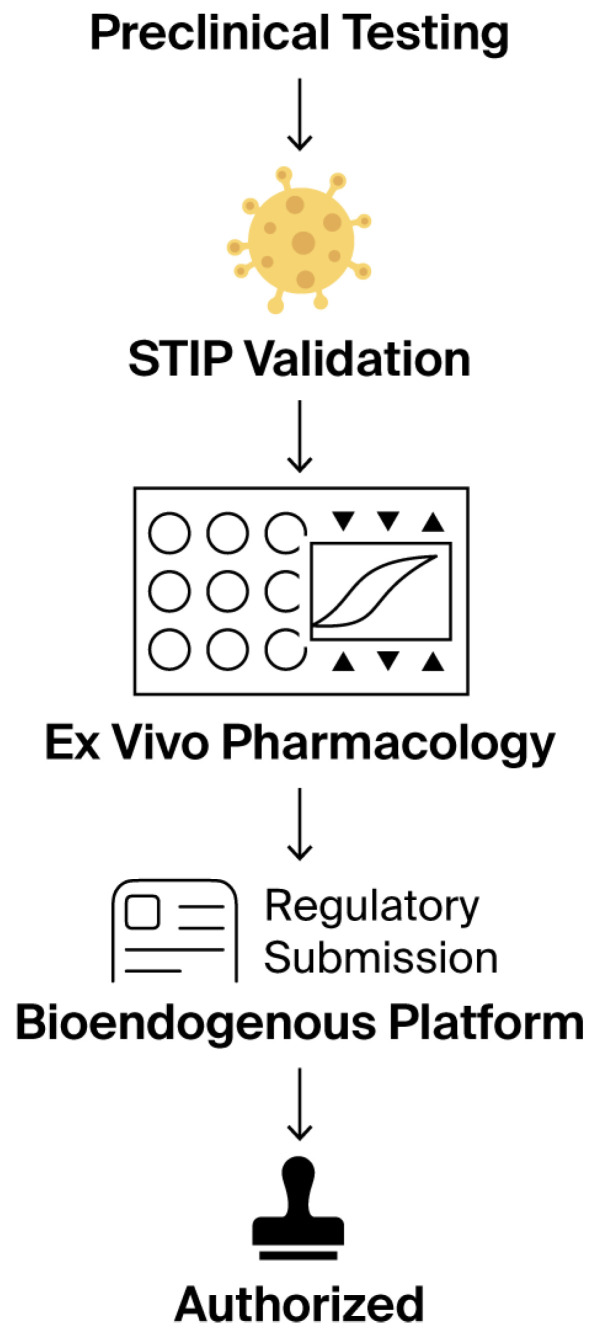
This stepwise regulatory flowchart outlines the non-clinical validation journey of a bioendogenous platform under the ICT-EP framework. The process begins with standard preclinical testing, transitions through STIP validation—where ex vivo phenotypic metrics, such as ΔC% and viability, are captured—then passes through ex vivo pharmacology using standardized, non-destructive assays. This structured evidence is compiled into a regulatory submission package that explicitly recognizes the product’s non-pharmacodynamic nature. The pathway concludes with authorization, emphasizing that approval is possible without systemic exposure, clinical endpoints, or human trials, when proportionate documentation is supplied through harmonized, reproducible non-clinical platforms. Symbols within the Ex Vivo Pharmacology panel represent assay wells (circles), experimental endpoints (triangles), and a representative response curve (center).

**Table 1 biomedicines-13-02101-t001:** Batch-level metrics summary.

Batch	ΔC%	AUC	Log-Phase Slope	Cumulative Viability	Secretomic Level (pg/mL)
FV-001	+6.8	3.25	0.024	6.4%	20.7
FV-002	+3.9	8.63	0.028	12.7%	135.2
FV-003	+3.2	10.12	0.072	16.7%	8.8
FV-004	+19.7	3.56	0.067	21.7%	18.4
FV-005	+22.7	6.48	0.072	6.4%	29.7

**Table 2 biomedicines-13-02101-t002:** Vesicle–cell interaction map (representative output view). Grid representation of vesicle–cell interaction outcomes in the kinetic profiling platform. Each entry denotes a single assay performed according to a standardized ex vivo protocol. Symbols correspond to generalized documentation categories applied during internal quality recording: ▲ indicates increased kinetic output, ▼ indicates decreased output, and — indicates no significant shift. These symbols are used exclusively as internal documentation markers and do not represent functional classification or therapeutic effect. No therapeutic inference or biological interpretation is assigned.

Vesicle	BEWO	A375	BEWO (rep)
FV-001	▲	▼	▼
FV-002	▼	—	▼
FV-003	—	▼	▼
FV-004	▲	▼	—
FV-005	▼	—	▼

**Table 3 biomedicines-13-02101-t003:** Multimetric output per vesicle–cell assay unit. Values are reported independently and not aggregated. The “Notes” column reflects procedural flags used internally, without interpretive weight.

Vesicle Fraction	Cell Line	Δ Confluence (%)	ΔT (h)	Death Signal (%)	IFN-γ/IL-10 Ratio	Notes
FV-001	BEWO	+34.1	10.2	1.1	2.1	Stable
FV-002	A375	−28.7	12.4	2.8	5.9	Stable
FV-003	MCF-7	+1.6	—	0.9	1.0	Low shift
FV-004	BEWO	+31.8	11.0	1.4	2.5	Stable
FV-005	A375	−29.5	13.1	2.6	6.1	Stable

**Table 4 biomedicines-13-02101-t004:** Cross-condition metric summary for internal documentation.

Metric	Range Observed	Inter-Batch CV%	Internal QC Threshold
Δ Confluence (%)	−29.5% to +34.1%	<12%	±5%
Divergence (ΔT, h)	10.2 to 13.1	<10%	2 h
Death Signal (%)	0.9% to 2.8%	<15%	3%
IFN-γ/IL-10 Ratio	1.0 to 6.1	<14%	n/a

**Table 5 biomedicines-13-02101-t005:** Comparison between clinical trial models and ex vivo functional protocols.

Criterion	Classical Clinical Trial	Ex Vivo Functional Protocol
Systemic absorption required?	Yes	No
Defined pharmacological dose?	Yes (e.g., Cmax, ED_50_)	No (non-systemic)
Therapeutic intent?	Yes	Not applicable
Generates adverse events?	Yes (monitored)	No (no systemic exposure)
Requires patient recruitment?	Yes	No (cell models only)
Uses placebo/randomization?	Yes	No
Provides functional evidence directly?	Indirect or endpoint-dependent	Yes (real-time metrics)
Document compatibility?	No (efficacy-based)	Yes (structure-function compatibility)
Allows batch comparability?	Indirect via PK/PD	Direct via reproducible metrics
CTD Module 5.3 integrable?	Only with human trial data	Yes (non-clinical documentation)
Reproducible without human subjects?	No	Yes

**Table 6 biomedicines-13-02101-t006:** Technical comparison: clinical trial vs. ex vivo functional documentation.

Criterion	Classical Clinical Trial	Ex Vivo Functional Protocol
Requires systemic absorption?	Yes	No
Defines pharmacological dose?	Yes (ED_50_, Cmax, LD_50_)	No (non-absorptive)
Seeks therapeutic effect?	Yes	No
Evaluates adverse events?	Yes	Not applicable
Requires human recruitment?	Yes (patients)	No (cell-based)
Uses randomization/placebo?	Yes	No
Provides real-time functional data?	Often indirect	Yes (curve, viability, cytokines)
Measures structural compatibility?	No	Yes
Enables batch comparability?	Indirect (via PK/PD)	Yes (technical replicates)
Compatible with CTD Module 5.3?	Only if trial is conducted	Yes (non-clinical evidence)
Reproducible without patients?	No	Yes

**Table 7 biomedicines-13-02101-t007:** Functional equivalence of evidence: drug trials vs. structural formulations.

Expectation	Clinical Trial (Drug)	Functional Documentation (Non-Drug)
Confirms batch consistency	Pharmacokinetics, patient arms	Cross-line kinetic and cytokine reproducibility
Documents product identity	Labeling, source control	Structural profiling, proteomic fingerprinting
Demonstrates mechanism of effect	Dose–response, receptor binding	Indirect through reproducible behavior
Validates safety	Adverse events, Phase I/II	Absence of toxicity, viability signal
Supports regulatory review	CTD Modules 2–5	CTD 5.3 (non-clinical, functional evidence)
Reproducibility across production cycles	Requires clinical lot tracking	Demonstrated via laboratory replicates

**Table 8 biomedicines-13-02101-t008:** Comparative documentation: clinical trials vs. functional phenotypic protocol.

Key Question	Clinical Trial (Drug)	Ex Vivo Documentation (Structural Platform)
How is safety confirmed?	Toxicity studies, maximum tolerated dose	Absence of cytotoxicity, viability signal consistency
How are off-target effects ruled out?	Adverse events in patients	Full phenotypic behavior tracked under standard protocol
How is batch consistency demonstrated?	Bioequivalence, therapeutic response	Reproducible kinetic and cytokine metrics across batches
How is reliability ensured?	Pharmacokinetics and efficacy over time	Divergence pattern stability and inter-batch reproducibility
Is it a new chemical entity (NCE)?	Assessed by toxicology and molecular profiling	Non-NCE with traceable proteomic identity
Is it appropriate for use in humans?	Confirmed by clinical outcomes	Technically stable and biologically neutral in cell models
Is it recognized by regulators?	Via NDA or similar programs	Acceptable under non-clinical documentation modules

**Table 9 biomedicines-13-02101-t009:** Functional mapping of CTD sections in non-pharmacodynamic products.

CTD Module or Section	Clinical Trial Contribution	Ex Vivo Functional Contribution
Module 5.3—Functional studies	Phases I–III, PK/PD, safety endpoints	Kinetics, viability, cytokine behavior, batch replication
2.7.3—Efficacy summary	Clinical effect, responder rate	Stability and divergence over time in cell models
2.7.4—Safety summary	Adverse events, side effects	No cytolysis or toxicity under test conditions
2.6—Non-clinical pharmacology	Animal models, systemic absorption	Localized compatibility, proteic profile stability
Module 3—Quality information	Analytical validation, identity, impurities	FTIR, DLS, document-linked traceability
1.12—Regulatory path justification	Rationale for omission or modification	Functional non-clinical file with supporting metrics

**Table 10 biomedicines-13-02101-t010:** Functional equivalence summary: clinical trials vs. ex vivo documentation models.

Regulatory Objective	Clinical Trial (Pharmacodynamic Drug)	Ex Vivo Functional Protocol (Non-Pharmacodynamic Formulation)
Demonstrates safety?	Yes, via adverse event tracking and toxicology	Yes, via absence of cytotoxicity, inflammatory signals, and death markers
Supports batch consistency?	Yes, through repeated trial arms and PK/PD trends	Yes, through kinetic, viability, and cytokine reproducibility
Provides functional data?	Yes, usually endpoint-based or indirect	Yes, with real-time kinetic curves and secretome response
Enables pre-use batch validation?	No, relies on post-release response	Yes, via standardized cell-line interaction profiles
Reproducible without patient data?	No, requires clinical enrollment	Yes, using validated sentinel lines under controlled conditions
Compatible with CTD Module 5.3?	Yes, as part of clinical data	Yes, under non-clinical functional documentation
Enables structural compatibility assessment?	Not as a primary endpoint	Yes, captured directly through ex vivo interaction
Recognized by regulators?	Yes, in standard therapeutic routes	Yes, in exempt or proportional documentation pathways

## Data Availability

Raw kinetic data, molecular profiles, and classification parameters are available from the corresponding author upon reasonable request. Access to these data may be subject to confidentiality agreements or material transfer conditions related to ongoing regulatory submissions. The full dataset is part of an active corporate editorial pipeline and is managed in accordance with contextual integrity and planned licensing frameworks.

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
