# Peer review of "Ex Vivo Traceability Platform for Phospholipoproteomic Formulations: Functional Evidence Without Clinical Exposure"

_biomedicines, 2025, doi:10.3390/biomedicines13092101_

Round 1

Reviewer 1 Report

Comments and Suggestions for Authors

This manuscript introduces a robust ex vivo platform designed to functionally validate biologically structured but non-systemically active formulations—namely phospholipoproteomic constructs—without the need for classical clinical trials. The system leverages kinetic profiling (e.g., ΔC, ΔT), viability assessments, and cytokine quantification (e.g., IFN-γ / IL-10 ratio) to provide documentation suitable for regulatory submission under CTD Module 5.3 and related frameworks. This work addresses a critical gap in the validation of structurally inert or non-pharmacodynamic products and aligns well with emerging regulatory needs.

Minor Revisions Suggested:

  1. Simplify Dense Jargon in Select Sections:
    While technically sound, some passages in Sections 3.4 and 4.6 are heavily loaded with regulatory terminology. Consider splitting longer sentences and using more intuitive phrasing to broaden accessibility to interdisciplinary readers.

  2. Clarify Product Applicability and Limits:
    Although the scope is broad, briefly clarifying the types of formulations best suited for STIP-based documentation (e.g., exosomes, liposomes, inert vesicular adjuvants) and explicitly stating which are excluded (e.g., pharmacodynamically active small molecules) would help delineate use cases.

  3. Supplementary Materials Accessibility:
    Ensure Supplementary Text S1 and Figures S0–S10 are available and referenced appropriately, especially since many mechanistic claims depend on these visual and technical assets.

  4. Highlight Clinical/Translational Use Case Example (Optional):
    A brief illustrative use case—such as a hypothetical product moving through the STIP pathway to submission—would increase practical relevance and highlight the pathway’s regulatory potential.

This manuscript makes a timely and valuable contribution to the field of regulatory science, particularly for structured formulations excluded from conventional pharmacodynamic trials. With minor editorial improvements and clarification of boundaries of applicability, this work will serve as a benchmark reference for non-interventional documentation strategies and harmonized regulatory submission design.

Author Response

Response Letter to Reviewer 1

Dear Editor,

We would like to thank Reviewer 1 for the careful evaluation of our manuscript entitled “Ex Vivo Traceability Platform for Phospholipoproteomic Formulations: Functional Evidence Without Clinical Exposure”. We appreciate the constructive comments, which have helped us to further improve the clarity and accessibility of the work. Below we provide a detailed point-by-point response.

Reviewer 1 – Comments and Responses

Comment 1: Simplify dense jargon in select sections (3.4 and 4.6)… consider splitting longer sentences and using more intuitive phrasing to broaden accessibility.

Response:
We thank the reviewer for this observation. Sections 3.4 and 4.6 have been revised to reduce regulatory jargon and long sentences. We adopted clearer phrasing while maintaining technical accuracy to ensure the text is more accessible to interdisciplinary readers.

Comment 2: Clarify product applicability and limits… briefly clarifying the types of formulations best suited for STIP and explicitly stating which are excluded.

Response:
We agree with this suggestion. A clarifying statement has been added in the Discussion (Section 4.2), emphasizing that STIP is particularly suited for non-pharmacodynamic vesicular or structural formulations (e.g., exosomes, liposomes, inert vesicular adjuvants) and is not designed for pharmacologically active small molecules or receptor-targeted drugs.

Comment 3: Ensure Supplementary Text S1 and Figures S0–S10 are available and referenced appropriately…

Response:
We appreciate the reminder. We have confirmed that Supplementary Text S1, Supplementary Table S1, and Supplementary Figures S0–S10 are included and appropriately cited in the main text. References to these materials were standardized across the manuscript.

Comment 4 (Optional): A brief illustrative use case—such as a hypothetical product moving through the STIP pathway to submission—would increase practical relevance.

Response:
Although optional, we considered this suggestion valuable. A concise illustrative example has been added in the Discussion (Section 4.6), showing how STIP outputs (ΔC%, viability, IFN-γ/IL-10 ratio) can be compiled into CTD Module 5.3 as a proportional evidentiary basis for regulatory review without requiring Phase I–III trials.

Closing

We thank Reviewer 1 again for the insightful comments. All requested changes have been incorporated, and we believe the revised manuscript has been significantly improved in clarity, scope definition, and accessibility.

Sincerely,

Dr Gutierrez-Sandoval
On behalf of all authors

Reviewer 2 Report

Comments and Suggestions for Authors

In this paper, the authors present the ex vivo traceability platform for phospholipoproteomic formulations. The proposed protocol is very strong from a technical and methodological perspective, especially with respect to regulatory compliance, reproducibility, and minimizing disruptions to the assay.

The authors obtained well-reproducible results; in particular, such metrics as final confluence shift, log-phase slope, and AUC showed low variability (<10–12% CV) for both stimulatory (e.g., BEWO, U87) and suppressive (e.g., A375, PANC-1) trajectories. This indicates that the analysis is technically stable and can be trusted for inter-batch and inter-line comparisons.

The proposed methodology can detect heterogeneous but consistent patterns; the system clearly distinguishes between stimulatory responses (early and sustained divergence, higher IL-6, low IFN-γ/IL-10 ratio), suppressive responses (reduced proliferation without cytotoxicity, high IFN-γ/low IL-10, high ratio), and neutral responses (minimal divergence, cytokine fluctuations without directional bias). This shows sensitivity to qualitatively different non-lethal interactions. Kinetics were measured in real time without terminal staining or destructive sampling, allowing cultures to be preserved for further analyses. The method is suitable for non-clinical screening, batch comparability, and regulatory audit trails, and can be integrated into CTD, SAP, or RWE frameworks.

The preprocessing workflow is well-designed for technical standardization, reproducibility, and regulatory documentation, which makes it good for high-throughput or audit-focused ex vivo studies. However, residual baseline variation and smoothing-related signal loss may reduce sensitivity to subtle biological effects. In subsection 2.5, the authors explicitly describe normalization to Tâ‚€ to reduce variability due to different starting confluences or initial cell densities. This addresses baseline differences to some extent. However, normalization to Tâ‚€ cannot completely remove intrinsic baseline variation (differences in metabolic state, cell-cycle phase, or stochastic growth fluctuations). For example, if a particular replicate starts with cells in a slightly different metabolic or cell-cycle state, its growth trajectory may deviate subtly from others, even after normalization.

The authors use the IFN-γ/IL-10 ratio, a widely accepted index of immune balance between proinflammatory Th1 activity and regulatory/anti-inflammatory signaling. Although this measure is useful as an orientation marker, it does not take into account the complexity of immune regulation. Complementary mediators such as IL-6, TNF-α, TGF-β, and IL-17, along with chemokine profiling, may provide a deeper insight into the functional dynamics of the immune response.

The authors determined that the cell lines MCF-7 and HepG2 exhibited minimal deviation from control trajectories and showed greater variability under conditions where no clear divergence was present. In such cases, the absence of a distinct divergence can mask subtle trends and complicate interpretation. For example, in cell lines MCF-7 and HepG2, such variability may arise from baseline fluctuations in growth kinetics, metabolic state, or cell-cycle distribution. This highlights the need for careful experimental design. Increasing the number of replicates, normalizing for baseline growth parameters, or incorporating orthogonal readouts (e.g., cell-cycle profiling or metabolic assays) may help disentangle true biological effects from inherent noise. Please provide a more detailed interpretation of this.

Author Response

Response Letter to Reviewer 2

Dear Editor,
We would like to thank Reviewer 2 for the careful and constructive review of our manuscript entitled “Ex Vivo Traceability Platform for Phospholipoproteomic Formulations: Functional Evidence Without Clinical Exposure”. We greatly appreciate the positive evaluation of the technical robustness and reproducibility of our protocol, and we address below the specific points raised.

Reviewer 2 – Comments and Responses

Comment 1: Residual baseline variation and smoothing-related signal loss may reduce sensitivity to subtle biological effects. Normalization to Tâ‚€ reduces variability but cannot fully account for intrinsic baseline differences (e.g., metabolic state, cell-cycle phase).

Response:
We thank the reviewer for this important observation. We have revised the Limitations section to explicitly acknowledge that normalization to Tâ‚€, while effective in reducing technical variability, cannot entirely compensate for intrinsic differences among replicates. These biological sources of variation (e.g., metabolic status or cycle distribution) are inherent to living systems and represent a limitation of any ex vivo platform. We now state this clearly in the manuscript.

Comment 2: The use of IFN-γ/IL-10 ratio is informative but does not fully capture immune complexity. Other mediators (IL-6, TNF-α, TGF-β, IL-17, chemokines) could provide deeper insights.

Response:
We agree with the reviewer. We have added a note in the Discussion acknowledging that while the IFN-γ/IL-10 ratio serves as a useful orientation marker, it represents only part of the immune regulatory spectrum. We now explicitly mention IL-6, TNF-α, TGF-β, IL-17, and chemokines as complementary mediators that could be integrated into future protocol expansions for a more comprehensive secretomic profile.

Comment 3: In MCF-7 and HepG2, minimal divergence from controls coincided with greater variability. The absence of distinct divergence may mask subtle trends. Suggestions: increase replicates, normalize baseline growth, or add orthogonal readouts (e.g., cell-cycle, metabolic assays). Please provide a more detailed interpretation.

Response:
We appreciate this valuable comment. We have expanded the Results and Discussion to provide a more detailed interpretation of MCF-7 and HepG2 findings. Specifically, we clarify that these “neutral” trajectories reflect both the absence of strong directional responses and a higher sensitivity to baseline fluctuations. We highlight that variability under non-divergent conditions is expected and must be interpreted with caution. In future applications, increasing the number of replicates and incorporating complementary readouts (cell-cycle or metabolic markers) will help refine interpretation. This recognition has been added both in the Discussion and Limitations.

Closing

We thank Reviewer 2 again for these constructive suggestions, which have helped us strengthen the clarity of our discussion and acknowledge the boundaries of the current approach. We believe the revised manuscript is now improved in interpretation, scope definition, and transparency.

Sincerely,
Dr Gutierrez-Sandoval
On behalf of all authors
